# Mechanical and Shape Memory Properties of Electrospun Polyurethane with Thiol-Ene Crosslinking

**DOI:** 10.3390/nano12030406

**Published:** 2022-01-26

**Authors:** Sam Briggs, Scott Herting, Grace Fletcher, Rachel Gruenbaum, Duncan J. Maitland

**Affiliations:** Biomedical Engineering, Texas A&M University, College Station, TX 77843, USA; sambriggs@tamu.edu (S.B.); smherting@gmail.com (S.H.); gracekfletch@tamu.edu (G.F.); rgruenbaum@tamu.edu (R.G.)

**Keywords:** shape memory, electrospinning, thiol-ene, crosslinking

## Abstract

The ability to treat complex medical issues often requires dynamic and versatile materials. Electrospinning is a fabrication technique which produces nano-/microfibers that can mimic the extracellular matrix of many biological tissues while shape memory polymers allow for geometric changes in devices upon implantation. Here, we present the fabrication of electrospun polyurethane which exhibits the shape memory effect. To improve the mechanical and shape memory properties of this system, we incorporate vinyl side chains in the polymer backbone which enable crosslinking via thiol-ene click chemistry post fabrication. We also discuss a novel technique to improve photoinitiated crosslinking for electrospun materials. A material with these properties is potentially beneficial for various medical applications, such as vascular anastomosis, and the characterization of this material will be valuable in directing those applications.

## 1. Introduction

Many medical procedures today require biomaterials that are adaptive in order to treat complex conditions. Recently, advances have been made in the areas of minimally invasive devices [1,2,3], stimuli responsive devices [4,5,6], and fabrication techniques [7,8,9]. The aim of this project is to combine the versatility of shape memory polymers (SMPs) with the benefits of electrospun fiber morphology. SMPs are often beneficial for medical applications as they allow for significant shape change in difficult to access areas [10]. For example, SMP foams have recently received Food and Drug Association approval as minimally invasive occlusion devices (IMPEDE Embolization Plug, Shape Memory Medical, Inc., Santa Clara, CA, USA). Electrospinning is an advantageous fabrication technique for tissue-contacting devices due to high porosity, flexibility, and extracellular matrix (ECM)-like microstructure [11]. Presented in this study is the fabrication and characterization of a crosslinkable, electrospun SMP with good mechanical and shape memory properties.

Shape memory polymers are a class of polymers that can undergo a geometric shape change in response to external stimulus such as temperature [12], pH [13], or light [14]. This shape memory effect occurs due to a polymer structure that includes switching segments and net-points [15]. Switching segments are stimuli sensitive, while the net-points act to remember the permanent shape. This shape memory ability makes these materials ideal for medical procedures as they can maintain one shape during delivery or implantation and then revert to a pre-programed permanent shape when stimulated, often by body heat, allowing for minimally invasive procedures. Polyurethanes are one of the most commonly used SMPs due to their toughness, durability, and biocompatibility [16,17]. Biomedical applications of SMP polyurethanes include vascular occlusion, clot removal, and vascular stenting [18,19].

Electrospinning is a relatively new fabrication technique in biomedical engineering. While the electrospinning process appeared in literature as early as the late 1800s [20], biomedical applications of the technique first appeared in the early 2000s [21]. In electrospinning, a thermoplastic polymer is either melted or dissolved in a solvent and then pumped through a needle to which a high voltage is applied. This builds up charge on the polymer which is then drawn towards a grounded collector forming what is called a Taylor cone at the needle. During the drawing process, the fiber whips back and forth allowing the solvent to evaporate off, resulting in the deposition of dry micro/nano fibers which can be collected into a mat. These fibrous materials are advantageous to biomedical applications because they often mimic the ECM of many tissues and have a high surface area with good flexibility [22]. Additionally, alterations in the fabrication method can generate a final material with anisotropic properties through fiber alignment. Applications for electrospun materials have a wide breadth, including drug delivery [23], tissue culture [24], and vascular devices [25].

One limitation of the electrospinning process is that it generally requires a thermoplastic material which can be dissolved in a solvent and extruded through a needle. This need for a thermoplastic limits the applications of electrospun materials as thermoplastic materials generally have lower tensile strength than thermosets and may be influenced by high temperatures or solvents used in many medical procedures (e.g., DMSO used for injection of liquid embolic agents for arteriovenous malformation embolization [26]). Additionally, for shape memory materials, thermosets have improved shape fixity and shape recovery over their thermoplastic counterparts [27]. Previous work in our lab developed a thermoplastic material with pendent vinyl side chains incorporated into the backbone which enabled post-fabrication crosslinking via thiol-ene click chemistry [28]. This allowed the material to be fabricated into the desired geometry via traditional thermoplastic processing techniques requiring melting or solvent dissolution, and then crosslinked post-fabrication for a final device with the advantages of thermosets. Electrospinning is a well-suited fabrication technique for this type of material as it also requires an initial thermoplastic material but can benefit from enhanced properties associated with crosslinking. Other groups have worked to utilize post fabrication crosslinking to achieve thermoset electrospun fibers [29]. Gong et al., utilized UV exposure to generate a chemically crosslinked poly(e-caprolactone) (PCL) with incorporated carbon nanotubes [30]. Zhang et al., also demonstrated the ability to crosslink electrospun PCL with gel fractions in the range of 80–90% [31]. However, to the best of our knowledge, no other group has demonstrated the benefit of crosslinking using thiol-ene crosslinking to achieve a material with improved gel fraction, tensile strength, and shape recovery.

Click chemistry is a valuable class of reactions described by Sharpless et al. [32] which allows for high yield reactions in ambient conditions without resulting in toxic byproducts. Thiol-ene click chemistry is a specific type of click chemistry which involves the radical mediated reaction of thiols with alkenes [33]. This photoinitiated chemistry is highly controllable and proceeds in atmospheric conditions. This has made it popular in polymer synthesis, including biomedical polymers. It has been used in photocurable dental resins [34,35], peptide science [36], and microfluidics [37]. For systems using thiol-ene click chemistry for polymer synthesis, one limitation is often the inability to decouple the glass transition T_g_ from the crosslinking density. The system presented here, where the thiol-ene reaction is used for crosslinking between polymer chains rather than for primary polymer synthesis, allows us to combine the mechanical properties of polyurethane systems with the versatility of thiol-ene click reactions.

Presented in this study is a polymer system which incorporates the benefits of thermoset materials and the electrospinning fabrication technique. Techniques for improving crosslinking effectiveness are demonstrated and mechanical and shape memory properties are characterized. This novel material has unique morphological, mechanical, and shape memory properties which make it a valuable addition to the materials toolkit in polymeric biomaterials.

## 2. Materials and Methods

### 2.1. Materials

Trimethylolpropane allyl ether (TMPAE), chloroform, N-N dimethylformamide (DMF), 2,2-dimethoxy-2-phenylacetophenone (DMPA), acetone, and tetrahydrofuran (THF) were purchased from Sigma-Aldrich Inc. (St. Louis, MO, USA) and used as received. 1,4-butanediol (1,4-BD) and pentaerythritol tetrakis(3-mercaptopropionate) (PETMP) were purchased from MilliporeSigma (Burlington, MA, USA) and used as received. Tetrakis(2,4-pentanedionato)zirconium(IV) (Zr Cat.) and trimethylhexamethylene diisocyanate (TMHDI) were purchased from TCI Chemical (Tokyo, Japan) and used as received.

### 2.2. Thermoplastic Synthesis

Thermoplastics were synthesized as previously described by Hearon et al., with some modifications. For synthesis, all components were added to a three-neck round bottom flask (RBF) containing a football-shaped stir bar under dry air in a LabConco glove box. Monomers (TMPAE, 1,4-BD, TMHDI) were added at an isocyanate to hydroxyl (NCO:OH) ratio of 1.02:1.00 with a final weight of 45 g. TMPAE constituted 10% of the OH molar concentration while 1,4-BD constituted the remaining 90%. Monomers were then diluted in 1.9 g THF/g monomer and 0.01% overall weight zirconium (IV) catalyst was added. The combined monomers, solvent, and catalyst were then transferred from the glove box to a fume hood, connected to a reflux condenser, and placed in an oil bath heated to 80 °C. Dried nitrogen was pumped into the RBF through a needle with a pressure relief needle on the top of the condenser to provide a nitrogen blanket during the reaction. The stir bar was set to 400 RPM to mix the components during the reaction. The reaction was allowed to proceed for 4 days until attenuated total reflectance Fourier transform infrared (ATR-FTIR) (Bruker ALPHA-Platinum, Bruker, Billerica, MA, USA) readings indicated no isocyanate functional groups remained. The reaction was then poured into treated glass Petri dishes and placed in a vacuum oven to dry. The samples were dried in the oven at 50 °C for 24 h at which point the temperature was raised to 80 °C for another 24 h. The samples were then kept in a vacuum oven for an additional 48 h to ensure they were completely dry. Molecular weight was determined using a TOSOH ambient temperature GPC (Tosoh Bioscience LLC, King of Prussia, PA, USA) where samples were dissolved at 1 mg/mL in THF, and a polystyrene standard was used.

### 2.3. Electrospinning Process

Solutions were prepared for electrospinning by dissolving the synthesized thermoplastic at a 1:4 weight concentration of Chloroform and DMF at a 3:1 ratio. This was mixed on an orbital shaker overnight to allow the thermoplastic to completely dissolve. TMPAE was then added at a 2:1 SH to C=C ratio along with 1.0 wt% DMPA photoinitiator. An excess of thiol was used because preliminary studies indicated PETMP may be lost to evaporation during the electrospinning process. Additionally, previous work with this material (Hearon et al.) demonstrated a decrease in material modulus when the SH:C=C ratio is less than one. This was allowed to mix for one hour before being loaded into a 5 mL syringe and placed in a syringe pump. The solution was pumped through a small diameter polytetrafluorethylene (PTFE) tube into the enclosed elctrospinning box at the rate of 1.5 mL/h to a blunt tip 18 G needle. A voltage of 25 kV was applied to the needle via a high voltage power supply (Gamma High Voltage Research Group Inc., Ormond Beach, FL, USA), and a ground was applied to a rotating mandrel collector with a diameter of 9 cm and a width of 15 cm. The gap distance between the dispensing needle and collector was 22 cm, and the rotation speed for the mandrel was controlled by a stepper motor and was set at 60 RPM. The electrospinning process continued until approximately 7 g of thermoplastic had been deposited. The thickness of the collected mat ranged from 0.20 to 0.35 mm.

### 2.4. Sample Crosslinking

UV crosslinking was performed in a UV oven box (IntelliRay 400, Uvitron, West Springfield, MA, USA). Samples were placed between two glass slides with 0.044 mm spacers during the UV process to prevent the warping of samples swollen in acetone. For samples swollen in acetone, the desired concentration of acetone was made by mixing acetone with RO water volumetrically. The solution was then mixed and added to the sample with a dropper. Approximately 10 mL of solution was added to each sample and samples were then placed between the glass slides and allowed to swell for 5 min before being placed in the UV oven. The UV oven was set to 50% intensity and samples were UV irradiated for the desired amount of time on both sides.

### 2.5. Scanning Electron Microscope (SEM) Imaging

Prior to imaging, samples were seated on a metal stub with carbon black tape and sputter coated using a Ted Pella Cressington 108 gold sputter coater (Ted Pella Inc., Redding, CA, USA). Sputter coating was performed for 60 s with a working distance of 3 mm to achieve a gold layer thickness of approximately 30 nm. SEM images were collected using a JEOL JCM-5000 Neoscope benchtop SEM (JEOL USA Inc., Peabody, MA, USA).

### 2.6. Gel Fraction Testing

Gel fractions for different treatments were determined using THF as the solvent. Three samples from each testing group were prepared and completely dried before an initial weighing after which they were placed in a clean glass vial. 20 mL THF was then added to the vial and the samples were incubated in an oven at 50 °C for 72 h with regular agitation. The THF was then decanted off and samples were dried in an oven at 50 °C with vacuum for 24 h. After drying, the samples were weighed again, and the gel fraction was calculated according to Equation (1)
(1)$$\mathrm{Gel}\;\mathrm{Fraction}\left(\%\right)=\frac{M^{\prime }}{M^{0}}\times 100$$
where *M*′ is the weight after exposure to THF and *M*^0^ is the weight before treatment.

### 2.7. Tensile Testing

Tensile testing was performed using an Instron 5965 (Instron Inc., Norwood, MA, USA). Samples were cut using a surgical scalpel into rectangles with a width of 7 mm and thicknesses ranging from 0.25 to 0.35 mm. The thicknesses of samples were measured using a digital micrometer rachet. The thickness of samples were randomized throughout the treatment groups to avoid any bias based on different thicknesses. The gauge length for the tests was 20 mm and the strain rate used was 10 mm/min. Three replicates were performed for each testing group.

### 2.8. Dynamic Mechanical Analysis Testing

Dynamic mechanical analysis (DMA) testing was performed using a DMA Q800 (TA Instruments, New Castle, DE, USA). For all tests, samples had a starting gauge length of 7 mm and a width of 6.4 mm and thicknesses ranging from 0.25 to 0.35 mm. For T_g_ testing, the instrument was run in multi-frequency strain rate mode with oscillating strain of 0.1% and oscillation rate of 1 Hz. Samples were cooled to −20 °C and held as isothermal for 10 min. The temperature was then ramped at a rate of 1.5 °C/min up to 120 °C. For recovery strain tests, the instrument was run in strain rate mode. Samples were heated to 60 °C and allowed to equilibrate for 10 min. They were then strained to 100% strain at a rate of 300%/min and held as isothermal for 5 min. The temperature was then lowered to −20 °C and held as isothermal for 5 min. The instrument force was then set to 0 N and temperature was jumped to 10 °C. The temperature was then ramped at the rate of 1 °C/min up to 45 °C for un-crosslinked samples and 60 °C for crosslinked samples and then held as isothermal for 30 min. For the cyclical shape recovery test, the samples were heated to 60 °C and allowed to equilibrate for 10 min. They were then strained to 100% strain at a rate of 300%/min and held as isothermal for 5 min. The temperature was then lowered to −20 °C and held isothermal for 5 min. The instrument force was then set to 0 N, and the temperature was jumped to 20 °C. The temperature was then ramped at a rate of 3 °C/min to 60 °C and held as isothermal for 20 min, after which, it was again strained to the original 100% strain and the process repeated. This cycle was repeated four times for a total of five cycles. Shape recovery strain was measured according to Equation (2):(2)$$Shape\;Recovery\;Strain\;\left(\%\right)=\frac{S-R^{n}}{S-R^{n-1}}\times 100$$
where *S* is the strain applied by the DMA, *R^n^* is the strain after recovery for cycle *n*, and *R^n^*^−1^ is the strain after recovery for cycle *n* − 1.

Force recovery tests were also run in strain rate mode. Samples were heated to 60 °C and allowed to equilibrate for 10 min. They were then strained to 100% and held as isothermal for 2 min. The temperature was then lowered to −20 °C and held as isothermal for 10 min. The control arm was locked in place, and the recovery force was measured as the samples were heated to 60 °C at a rate of 2 °C/min. All tests were performed in triplicate for each sample with exception of the no crosslink sample in the strain recovery test for which only one sample could be successfully completed due to material softness after heating.

### 2.9. Macroscale Shape Memory Testing

To demonstrate the shape memory effect of the electrospun material in physiologically relevant conditions, testing was performed in 37 °C water. A 30% acet, 30 min sample 20 mm in length was heated to 60 °C and stretched to approximately 40mm in length and allowed to cool back to room temperature. It was then hung vertically in a glass beaker and submerged in 37 °C water to induce shape recovery.

### 2.10. Statistical Analysis

All statistical analysis was performed in JMP where an unpaired Student’s *t*-test with *p* < 0.05 was used to create connecting letter reports. For the connecting letters report, shared letters between the samples indicate no statistical differences, while a lack of shared letters indicates statistical difference.

## 3. Results

### 3.1. Thermoplastic Synthesis

Figure 1 presents a schematic that is representative of the chemistry used in this study which incorporates vinyl side chains, allowing for thiol-ene crosslinking post fabrication. Molecular weights from the polymer used in the study are also presented. Figure 2 shows FTIR data gathered during the polymerization process. Reduction of the isocyanate peak at 2265 cm^−1^ can be observed along with typical polyurethane peaks at 3300 and 1715 cm^−1^.

### 3.2. Electropsinning and Crosslinking via Swelling

Preliminary studies with electrospinning the thermoplastic allowed us to determine the parameters to successfully generate non-beaded fibers in a consistent manner. However, early investigation revealed that thiol-ene crosslinking was inefficient for samples UV crosslinked directly after electrospinning. However, it was found that the samples could be swollen in dilute concentrations of acetone directly before UV exposure to achieve much higher crosslinking efficiency. Figure 3 shows the influence of swelling on fiber morphology after fibers were allowed to dry. The images demonstrate that at 35% acetone concentration, the fibers began to meld and lose their definition, which would alter the microstructure and porosity of the final material. However, the images in Figure 3 demonstrate that below this threshold, the final fiber morphology is not altered by the swelling process. This guided down-selection of concentrations used in future studies. Table 1 outlines the treatments that were tested in both tensile and DMA testing. This testing allowed us to determine the influence of acetone concentration and UV exposure time on the properties of these materials. Table 1 also shows the gel fraction for each of the conditions tested. Gel fraction testing was the first indicator of the benefit to crosslinking achieved by swelling the polymers.

### 3.3. Tensile Testing

Tensile testing was used to further distinguish the influence of crosslinking parameters on material properties. Figure 4a shows the stress/strain curves for all conditions tested. Figure 4b shows that crosslinking reduces the elongation at break for the materials while Figure 4c,d show that it increases the ultimate tensile strength (UTS) and Young’s modulus. Additionally, a distinct jump in strength for materials crosslinked by swelling was observed. In comparing the influence of acetone concentration, it was observed that samples treated with 30% acetone had statistically higher UTS than those treated with 10%. Furthermore, a distinct increase in modulus was observed as the UV exposure time for the samples increased.

### 3.4. Dynamic Mechanical Analysis

DMA testing was used to further understand the influence of crosslinking on the shape memory properties of this material. For these tests, three sample groups were used: No Crosslink, No Swell, and 30% Acet, 30 min. Figure 5 shows the tan delta curves for these three samples, demonstrating a shift in glass transition temperature with increased crosslinking as well as improved mechanical stability of the materials above their T_g_.

Figure 6 shows the storage modulus of these materials during a temperature sweep encompassing the glass transition temperature for each material. The No Crosslink sample group was unable to maintain structural integrity above 55 °C, and thus, no modulus below T_g_ could be recorded. The No Swell sample group was able to maintain structural integrity up to 120 °C but continued to decrease in storage modulus above its T_g_. The 30% Acet., 30 min sample group had the highest modulus of any group both above and below its T_g_ and maintained its modulus above its T_g_.

Recovery strain testing was performed for each sample using DMA, and results are shown in Figure 7. Due to their instability at high temperatures, the No Crosslink samples were only heated to 45 °C rather than the 60 °C like the No Swell and 30% Acet, 30 min samples. Interestingly, even though 30% Acet, 30 min samples had the highest glass transition temperature, they had the earliest shape recovery onset point. Additionally, they had a much faster recovery rate which was measured as the temporal distance between the onset points for the beginning and ending of the shape recovery process. The swollen crosslinked samples also had the highest shape recovery, suggesting that efficient crosslinking was an important factor in shape memory properties.

A cyclical shape recovery test was run on 30% Acet., 30 min samples and is presented in Figure 8. This test demonstrates that while the shape recovery strain for the first cycle was 85.62%, the shape recovery strain of subsequent cycles was nearly 100%.

Figure 9 shows the shape recovery force for these materials. For these samples, the recovery force was measured as the difference in stress between the low point and the stress after shape recovery, while the post-recovery force was measured as the maximum force achieved after shape recovery. A distinct increase in the base stress along with the recovery stress of swollen and crosslinked samples can be observed.

Figure 10 serves to demonstrate the shape memory effect of this material at physiologically relevant temperatures of 37 °C and in an aqueous environment. The test shows that the material recovers quickly in 37 °C water and achieves similar recovery strains to that shown in the DMA testing in Figure 7.

## 4. Discussion

Chemical, mechanical, and thermal characterization is a vital step in determining the potential uses of any material. In this work, we have thoroughly investigated the properties of an electrospun shape memory polyurethane with pendent vinyl groups which enable thiol-ene crosslinking post fabrication. This will inform future work in utilizing this material for biomedical applications.

Our first step in understanding this material was to characterize the bulk properties of the thermoplastic. GPC testing showed that we could synthesize a fairly low molecular weight polyurethane, which could be easily dissolved for electrospinning. ATR-FTIR demonstrated efficient reaction of the polymer. Because this was a custom polymer synthesized in-house, initial work was required to determine the parameters required to efficiently electrospin the material. A DOE approach allowed us to determine parameters, which generated fibers in the 1–10 µm range. However, it quickly became apparent through gel fraction testing that crosslinking did not occur efficiently for the electrospun samples. In this work we utilize a catalyzed free-radical addition thiol-ene click reaction to crosslink the samples. This requires that our photoinitiator (DMPA), polythiol (PETMP) and alkene (TMPAE component of polymer backbone) all be in appropriate proximity for the thiol-ene reaction to occur. Previous work with this polymer system by Hearon et al., demonstrated that efficient crosslinking could occur with small amounts of solvent; however, crosslinking was never characterized for samples that had been completely dried. Additionally, Xu et al. [38] showed that aggregation of nanoparticles can occur during the electrospinning process. We hypothesized that the inefficient crosslinking was either due to the lack of solvent in the resultant fibers inhibiting the thiol-ene click reaction or aggregation of PETMP or DMPA in the polyurethane fibers leading to poor distribution. For this reason, electrospun mats were crosslinked while swollen in dilute acetone solvents. It was hypothesized that this would help with molecular motion to improve thiol-ene reaction efficiency and allow any small molecules such as PETMP or DMPA to redistribute in the polymer network. Results show that this treatment did indeed have a distinct effect on the crosslinking efficiency with increases in gel fraction along with mechanical and shape memory properties.

Tensile testing was used to investigate the influence of crosslinking under various parameters on mechanical strength. It was observed that UV exposure under any condition lead to an approximately 37% reduction in elongation in break. This suggests that even though the non-swollen samples had much lower crosslinking efficiency than swollen samples, it was still enough to restrict polymer chain movement during elongation. UTS measurements showed the drastic difference between swollen and non-swollen samples with an approximate five fold increase in UTS for swollen samples. Looking at the Young’s modulus also showed the influence of crosslinking, where all crosslinked samples had significantly higher Young’s modulus than the no crosslink samples. The influence of differing swelling parameters was also demonstrated, showing that acetone concentration does not have a significant influence on modulus as the values for samples treated with 10% and 30% acetone were similar. However, UV exposure time does appear to have a significant impact with a steady increase in the modulus as the exposure time was increased from 1 to 10 to 30 min.

The testing performed on the DMA provided a better understanding of the thermomechanical and shape memory properties of samples with different crosslinking treatments. Temperature sweep measurements showed that as the crosslinking was increased, there was a corresponding increase in T_g_ and storage modulus above the T_g_. Macroscale shape memory testing in 37 °C water shows that the material will actuate quickly under physiological conditions. Additionally, if the biomedical application is in dry conditions, a small amount of warm saline could be used to achieve material actuation. Shape recovery testing demonstrated not only an increase in the amount of strain that could be recovered in swollen samples, but also a dramatic increase in the rate or recovery for those samples. This is an important parameter when considering use of this material in biomedical applications as it will determine how quickly the material responds when implanted in the desired location. Cyclical strain recovery testing run on the 30% Acet., 30 min samples showed that after the first cycle, the materials would recover 99%+ of the strain. This indicates that after the molecular stresses induced by the electrospinning and crosslinking processes were released by the first cycle, the recovery of the material could be predicted very accurately. Finally, shape recovery force testing demonstrated that the swollen samples not only maintained higher stresses during a temperature sweep, but also exerted more force during the shape recovery process. The differences in shape memory properties between crosslinked and uncrosslinked samples is caused by a difference in recovery mechanism [39,40]. For uncrosslinked samples, physical hydrogen bonding in the hard segments of the polyurethane act as the netpoints which anchor the shape memory. The crosslinked samples have the addition of covalent thiol-ene linkages which act as much stronger netpoints. Due to these linkages, covalent samples have high recovery strains (Figure 7) and recovery force (Figure 9). The more durable covalent bonds also enable the material to have stable cyclical recovery as shown in Figure 8.

## 5. Conclusions

The goal of this study was to determine the material properties of this system in order to inform potential biomedical applications. An electrospun material with high gel fraction, good tensile strength, and impressive shape recovery may be useful in medical procedures such as vascular, tendon, or bone repair. Application of electrospun materials in vascular grafts and repair has been a popular area of research [25,41,42,43,44]; however, none have incorporated the shape memory effect, which could improve sleeve fitting and reduce tension at the healing site for anastomosis applications. These studies will help direct future studies into the biomedical uses of this material.

## Figures and Tables

**Figure 1 nanomaterials-12-00406-f001:**
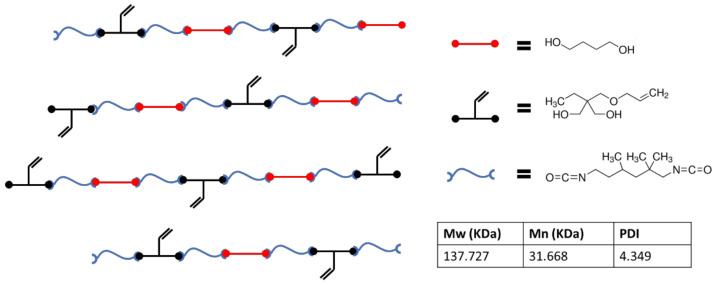
Schematic representing the thermoplastic polymer chemistry utilized during this study along with GPC data on the synthesized polymer.

**Figure 2 nanomaterials-12-00406-f002:**
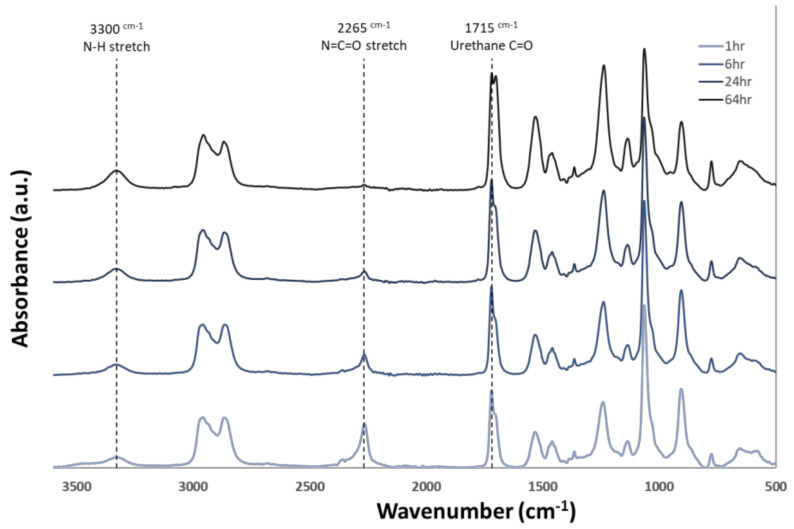
Attenuated total reflectance Fourier transform infrared (ATR − FTIR) spectra of thermoplastic over the duration of the synthesis process. Reduction of the isocyanate peak at 2265 cm^−1^ and an increase of urethane linkage groups at 1715 cm^−1^ and 3300 cm^−1^ can be seen as the reaction proceeds.

**Figure 3 nanomaterials-12-00406-f003:**
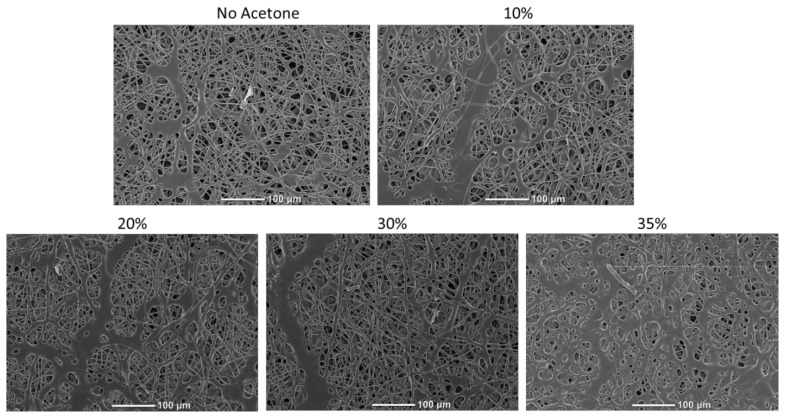
Scanning electron microscopy (SEM) images of electrospin scaffolds which were swollen with varying concentrations of acetone and then dried. Melding of fibers was observed when 35% acetone was used.

**Figure 4 nanomaterials-12-00406-f004:**
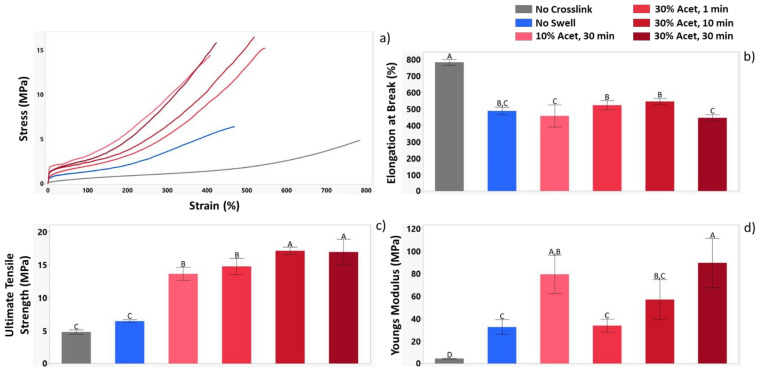
Tensile data for each of the sample groups demonstrating the influence of crosslinking parameters on mechanical properties. Results presented include (**a**) Representative stress-strain curves (**b**) elongation at break (**c**) ultimate tensile strength (UTS) and (**d**) Young’s modulus. n = 3 for all conditions tested. For the letters used on the bar graphs in (A–D), shared letters between the samples indicate no statistical differences, while a lack of shared letters indicates a statistical difference.

**Figure 5 nanomaterials-12-00406-f005:**
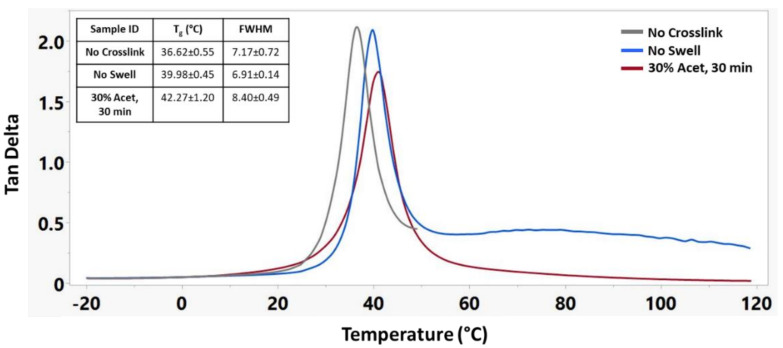
Representative tan delta curves of different crosslinking treatments along with corresponding T_g_’s and full width half maximum (FWHM) measurements. n = 3 for each sample group.

**Figure 6 nanomaterials-12-00406-f006:**
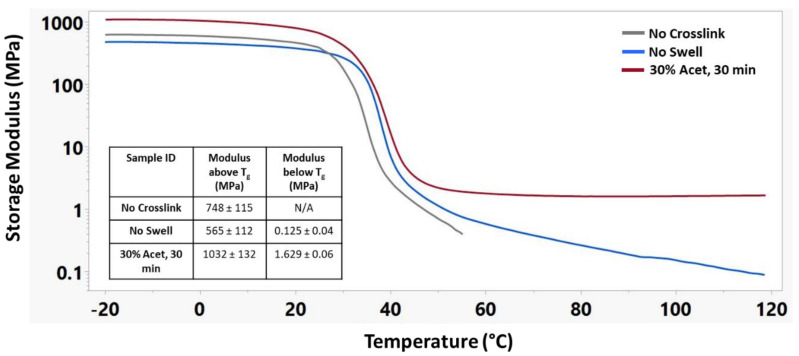
Representative storage modulus curves of different crosslinking treatments along with corresponding modulus above and below T_g._ n = 3 for each sample group.

**Figure 7 nanomaterials-12-00406-f007:**
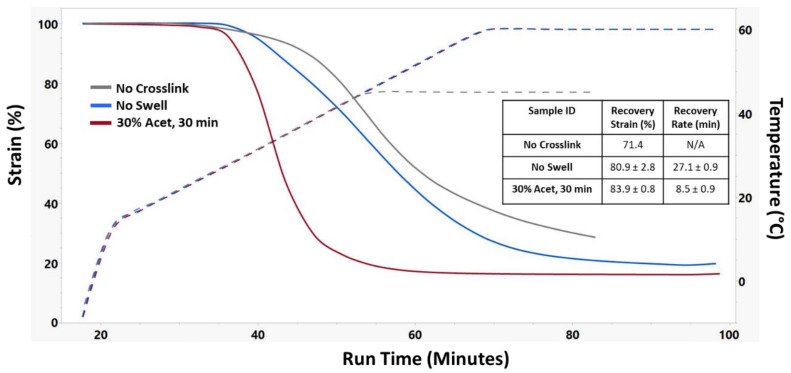
Representative shape recovery strain curves of different crosslinking treatments along with corresponding recovery strain and recovery rate. n = 3 for No Swell and 30% Acet, 30 min samples.

**Figure 8 nanomaterials-12-00406-f008:**
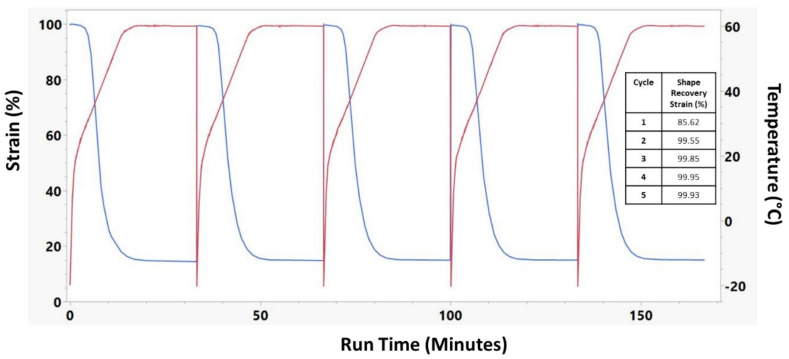
Cyclical shape recovery of 30% Acet, 30 min crosslinked electrospun sample. 5 full shape recovery cycles were performed. Red lines indicate the temperature while blue lines indicate the sample strain.

**Figure 9 nanomaterials-12-00406-f009:**
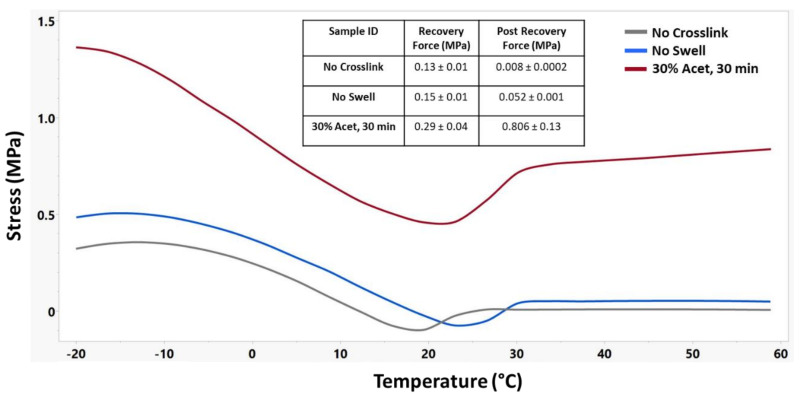
Representative shape recovery force curves of different crosslinking treatments along with corresponding recovery force and post recovery force. n = 3 for each sample group.

**Figure 10 nanomaterials-12-00406-f010:**
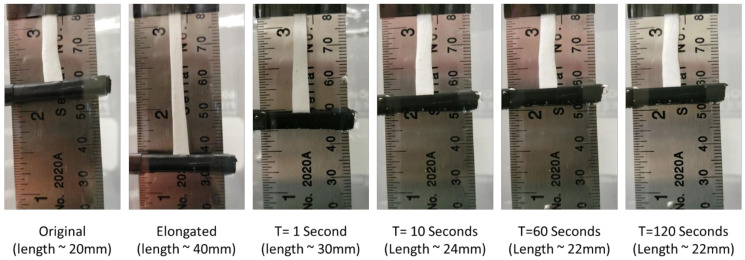
Macroscale shape recovery of 30% Acet, 30 min electrospun samples in 37 °C water. Images show that material recovers rapidly in body-temperature water and achieves approximately 90% shape recovery.

**Table 1 nanomaterials-12-00406-t001:** Descriptions and sample names used throughout the study along with gel fraction results for each of those samples. n = 3 for gel fraction testing.

Sample ID	Sample Description	Gel Fraction (%) ± SD
No Crosslink	Electrospun sample with no crosslinking	N/A
No Swell	Electrospun sample exposed to 30 min of ultraviolet (UV) irradiation without Swelling	<1
10% Acet, 30 min	Electrospun sample exposed to 30 min of UV irradiation after swelling in 10% Acetone	91.0 ± 1.38
30% Acet, 1 min	Electrospun sample exposed to 1 min of UV irradiation after swelling in 30% Acetone	93.7 ± 0.80
30% Acet, 10 min	Electrospun sample exposed to 10 min of UV irradiation after swelling in 30% Acetone	94.6 ± 0.73
30% Acet, 30 min	Electrospun sample exposed to 30 min of UV irradiation after swelling in 30% Acetone	93.8 ± 0.99

## Data Availability

Data can be available upon request from the authors.

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
