# Peer review of "Mechanical and Shape Memory Properties of Electrospun Polyurethane with Thiol-Ene Crosslinking"

_nanomaterials, 2022, doi:10.3390/nano12030406_

Round 1

Reviewer 1 Report

This work attempts to the fabrication of electro spun polyurethane which exhibits the shape memory effect. To improve the mechanical and shape memory properties of this system, vinyl side chains were introduced into the polymer backbone which enable crosslinking via thiol-ene click chemistry post fabrication. The idea of this work is interesting, the author has done enough data to show the effect of material. However, there are some issues that cannot be ignored in the article that need to be further revision and then resubmit for journal. The issues are listed as follows:

  1. Please write the full name of the FDA in the first paragraph. 
  2. Is there an intuitive experiment can be proved the shape memory effect of the material?
  3. The article should add a conclusion chapter.
  4. The author said it potentially beneficial for various medical applications, if possible, please add relevant experiments.
  5. The article has no in-depth mechanism discussion.

Reviewer 2 Report

In this manuscript, the authors presented a possible way for preparing ECM-mimic nano/microfibers based on shape memory polyurethane. The knowledge about the electrospun polyurethane with shape memory capability is there already since some years. However, the authors in this study demonstrates the detailed preparation and characterization of an electrospun polyurethane with thiol-ene crosslinking. In my opinion, the manuscript is worth to be published in Nanomaterials, once the following issues are clearly addressed.

  1. The authors only used specific SH : C=C feed ratio of 2 : 1 to crosslink the electrospun PU. Please provide an appropriate explanation for why the authors applied this ratio.

  1. The swelling method was found to achieve effective crosslinking of electrospun PU, however, it can also affect the microstructure and porosity of the electrospun and crosslinked scaffold. Supporting data and discussion for the effect of swelling on microstructure and porosity of the scaffold should be included in the revised manuscript.

  1. The shape memory transition temperature (~42 oC) of electrospun and cross-linked PU seems to be somewhat higher than that of body temperature. Moreover, in Figure 8, the recovery temperature of 60 oC was applied to test its cyclic shape memory characteristics. How can the authors control the transition temperature to be appropriate for the actual biomedical applications?
